# Evaluation of the PGPR Capacity of Four Bacterial Strains and Their Mixtures, Tested on *Lupinus albus* var. Dorado Seedlings, for the Bioremediation of Mercury-Polluted Soils

**Daniel González, Carlota Blanco, Agustín Probanza** 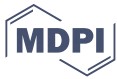**, Pedro A. Jiménez** and **Marina Robas ***

Department of Pharmaceutical Sciences and Health, Faculty of Pharmacy-Montepríncipe Campus, CEU San Pablo University, CEU Universities, Ctra. Boadilla del Monte Km 5.300, 28668 Boadilla del Monte, Spain; daniel.gonzalezreguero@ceu.es (D.G.); c.blanco17@usp.ceu.es (C.B.); a.probanza@ceu.es (A.P.); pedro.jimenezgomez@ceu.es (P.A.J.)
* Correspondence: marina.robasmora@ceu.es; Tel.: +34-620-25-81-84

**Abstract:** Soil contamination by mercury, which is one of the most toxic heavy metals due to its bioaccumulative capacity, poses a risk to the environment as well as health. The Almadén mining district in Ciudad Real, Spain is one of the most heavily-polluted sites in the world, making the soils unusable. Bioremediation, and more specifically phyto-rhizoremediation, based on the synergistic interaction established between plant and Plant Growth Promoting Rhizobacteria (PGPR), improves the plant's ability to grow, mobilize, accumulate, and extract contaminants from the soil. The objective of this study is to evaluate the plant growth-promoting ability of four PGPR strains (and mixtures), isolated from the bulk soil and rhizosphere of naturally grown plants in the Almadén mining district, when they are inoculated in emerged seeds of *Lupinus albus*, var. Dorado in the presence of high concentrations of mercury. After 20 days of incubation and subsequent harvesting of the seedlings, biometric measurements were carried out at the root and aerial levels. The results obtained show that the seeds treatment with PGPR strains improves plants biometry in the presence of mercury. Specifically, strain B2 (*Pseudomonas baetica*) and B1 (*Pseudomonas moraviensis*) were those that contributed the most to plant growth, both individually and as part of mixtures (CS5 and CS3). Thus, these are postulated to be good candidates for further in situ phyto-rhizoremediation tests of mercury-contaminated soils.

**Keywords:** PGPR; bioremediation; phyto-rhizoremediaton; mercury; soil contamination

## 1. Introduction

The Almadén mining district in Ciudad Real, Spain is an area of approximately 300 km$^2$, which is of geological interest worldwide, due to the fact that together with the Idrija mine in Slovenia [1], it is considered to have one of the largest deposits of mercury (Hg) along with the presence of high geogenic levels of Hg. The concentrations of this heavy metal in the soils of Almadén exceed 10$^6$ µg/kg [2].

More than 30% of the total Hg amount that has been obtained worldwide has come from the Almadén mines [3]. The majority of this production has been extracted from the main mine in the district, the Almadén mine, whose exploitation dates back to Roman times with more than 2000 years of history. Ultimately, campaigns against the use of Hg in Europe resulted in the permanent closure of the mine's operations in 2001 and the cessation of metallurgical activity in 2003. Currently, the area is maintained as a tourist attraction [4].

Elemental Hg (Hg$^0$) is a heavy metal with a silvery-white color that has been used in the medical and industrial fields for many years. This metal has been used in insecticides, dyes, and protectants for wood. Currently, its use is administered at the European level by Regulation (EU) 2017/852 on mercury and, since 2013, following the legally binding UN Convention of Minamata, it has been recognized as a global pollutant [5]. Hg pollution

can come from natural sources, such as volcanic emissions, or from anthropogenic origins, such as those resulting from certain industrial processes [6]. From these emissions, Hg enters the atmosphere in the form of $Hg^0$ vapor, where it can remain for up to 1.7 years [7]. Through diverse geological processes, Hg is deposited in the biosphere, where it can form organic and inorganic salts. In general, these compounds tend to remain in the aqueous phase as undissociated molecules with relatively low solubility values [6]. Bearing all of this in mind, it is considered that the environmental impact of Hg is significant, since it affects surface and groundwater, air, soils, and the biosphere as well [8].

From a toxicological point of view, it is a toxic metal, or in other words, it does not have a specific biological role, which means that its incorporation into the body is not necessary. Therefore, at certain doses it produces adverse effects such as problems in the development, growth, and reproduction of living beings [9].

The cessation of mining and metallurgical activity in the Almadén mining district has had socioeconomic consequences for the population [10]. This situation has made it necessary to establish new uses for the land affected by the high concentration and provide a solution to the problem of pollution; different options have been proposed for recovering these soils. Physicochemical methods are the most widely used, though not the most favorable, due to their aggressiveness toward the environment and high costs [11]. Nowadays, from a biotechnological point of view, the most promising options for soil decontamination are bioremediation processes based on the use of living organisms (plants, fungi and bacteria) to degrade, transform or eliminate toxic compounds into harmless or less toxic metabolic products. Strategies based on the use of microorganisms and their enzymes are of increasing interest for biotechnological applications [12] due to their lower costs and lower environmental impact [13]. However, more studies are needed regarding the microbial diversity of sites contaminated with heavy metals, since those are the places where strains that are more well-adapted with greater capabilities can be identified for use in the bioremediation of these spaces.

Within the field of bioremediation study, it is worth highlighting phytoremediation, which uses the ability of plants to extract soil pollutants [14]. The effect is much stronger and more efficient when the action of the plant is combined with the activity of the bacteria present in its rhizosphere. In this sense, it is worth highlighting the so-called Plant Growth Promoting Rhizobacteria (PGPR), highly efficient to increase plant growth and increase their tolerance to biotic and abiotic factors. This particular type of phytoremediation using PGPRs in combination with plants is called phytorhizoremediation [15]. The success of this technique lies in the fact that the rhizosphere is an interface in which plants and microorganisms establish complex and varied molecular relationships, which involve the nutrients transfer, as well as specific interactions mediated by the release of signaling molecules from plant roots [16].

In the present study, as a preliminary phase prior to field trials using phytorhizoremediation in situ in the mining district of Almadén, biological tests were carried out in controlled conditions with seeds and seedlings of the *Lupinus albus* var. Orden Dorado and pre-selected isolated strains based on PGPR capabilities and their Bio-Mercury Remediation Suitability Index (BMRSI). This index provides a comprehensive assessment of the appropriateness of the strain for its successful implementation in phytorhizoremediation tests [17]. It measures the bioremediation potential of the strain by the inclusion of different PGPR activities, such as indoleacetic acid production (IAA), ACC degradation capacity (via ACC deaminase; ACCd), siderophores production (SID), ability to solubilize phosphates ($PO_4^{3-}$) and the maximum bactericidal concentration (MBC) of Hg, which were all measured in vitro.

The genus *Lupinus*, known as lupine, is a plant with an extracting capability for heavy metals [18]. The choice of *Lupinus albus* var. Orden Dorado was made based on its ecophysiological characteristics of strong adaptability to the conditions that occur in mines; high salinity, excessive nitrates, and a low amount of nutrients [19]. The ability to solubilize and absorb soil elements thanks to extremely robust root development makes

this plant a candidate for remediation tests in this type of environment. Studies done by Quiñones et al. in 2013 and 2018 used the rhizobacteria—*Lupinus albus* model to study the capacity of this symbiotic pair to tolerate and/or accumulate Hg. They demonstrated that the inoculation of lupine plants with Hg-tolerant strains had a clear effect on their response to exposure to this heavy metal [20,21].

This study aimed at verifying whether the inoculation of *Lupinus albus* var. Orden Dorado by PGPR strains (and their mixtures) in soils contaminated with Hg produces greater plant development, and to select the best plant growth-promoting strains for subsequent in situ tests for the phyto-rhizoremediation of soils contaminated with Hg.

## 2. Materials and Methods

### 2.1. Bacterial Strains and Mixtures Tested

The strains tested in the present study were isolated from the plant's rhizosphere and bulk soil of "Plot 6" of the mining district of Almadén in Ciudad Real, Spain [18]. The PGPR capacity in the presence of Hg of four bacterial isolates (Table 1), and six mixtures, originating from the combination of the individual isolates (Table 2), was analyzed. These four strains were selected based on their values according to the Bio-Mercury Remediation Suitability Index (BMRSI), as well as on the study of auxin production (IAA), the presence of the enzyme 1-aminocyclopropane-1-carboxylate decarboxylase (ACCd), and the production of Siderophores (SID). Phosphate solubilization was also considered in this index, although our strains did not display this capability [17]. BMRSI can be calculated by using the following formula, where the values 1 and 0 for ACCd and $PO_4^{3-}$ indicate Presence and Absence, respectively:

$$BMRSI = [IAA (\mu g/mL) + ACCd (1/0) + SID (cm) + PO_4^{3-} (1/0)] + [MBC\ Hg (\mu g/mL)]$$

**Table 1.** Evaluation of the PGPR and BMRSI ability of the tested bacterial isolates.

| Strain | [HgCl2] ppm | [IAA] ppm | ACCd (p/a) | Siderophores (cm) | BMRSI (s.u.) | Strain Origin | 16s rRNA Identification |
|--------|-------------|-----------|------------|-------------------|--------------|---------------|-------------------------|
| A1 | 140 | 6.29 | − | 2.8 | 9.24 | *A. sativa* | *Brevibacterium frigolitolerans* |
| A2 | 140 | 6.16 | + | 0.0 | 7.30 | SL | *Bacillus toyonensis* |
| B1 | 140 | 7.06 | − | 0.8 | 8.00 | SL | *Pseudomonas moraviensis* |
| B2 | 160 | 7.85 | − | 0.0 | 8.00 | *A. sativa* | *Pseudomonas baetica* |

[HgCl$_2$]: maximum bactericidal Hg concentration (ppm); IAA: Indoleacetic acid production (ppm); ACCd (ACC deaminase): presence (+)/absence (−); Siderophores: measurement of the halo produced around the bacterial growth zone (cm); BMRSI (Bio-Mercury Remediation Suitability Index): unitless; Origin of the strain: SL (bulk soil), *A. sativa* (rhizosphere of *Avena sativa*).

**Table 2.** Mixtures formed from the strains in Table 1.

| Mixtures | CS1 | CS2 | CS3 | CS4 | CS5 | CS6 |
|----------|-----|-----|-----|-----|-----|-----|
| Strains | A1 + B1 | A1 + A2 | A1 + B2 | B1 + A2 | B1 + B2 | A2 + B2 |

### 2.2. Tested Plants

Seeds of the *Lupinus albus* var. Orden Dorado from the bank of the Extremadura Scientific and Technological Research Center were used.

### 2.3. Substrates

Two types of substrates were used: bulk soil from "Plot 6" of the Almadén mining district and sterile vermiculite. The latter was tested to assess the effects of Hg on the tested strains, beyond the shielding effect that the soil microbial communities may provide. The characteristics of the different substrates and treatments were the following:

- Contaminated soil, high Hg concentration: sample from "Plot 6" in Almadén, specifically from "The mine on the southern slope of Cerro Buitrones" [22]. The Hg concen-

tration in this plot was 1710 mg/kg total Hg, 0.609 mg/Kg soluble Hg and 7.3 mg/Kg (8 ppm) interchangeable Hg.

- Control soil, low Hg concentration: sample from "Plot 2" in Almadén, known as Fuente del Jardinillo [18]. This plot had a concentration of 5.03 mg/kg of total Hg: 0.0417 mg/kg of soluble Hg and 0.285 mg/kg of exchangeable Hg.
- Vermiculite without Hg: vermiculite is an inert substrate with a neutral pH, which is used in hydroponic crops.
- Vermiculite with Hg: a solution of 8 ppm of Hg was added to this substrate, to recreate the growth conditions of the plant in the soil subjected to high concentrations of Hg, typical of the study area (samples from "Plot 6").

### 2.4. Seed Pre-Germination

As a preliminary step, the seeds were imbibed in tap water at 4 °C for 24 h. Then, they were surface sterilized with three washes with 70% ethanol for 30 s. PVC trays (40 cm × 35 cm) were used for pre-germination, filled with sterile vermiculite brought to field water capacity with sterile tap water. Subsequently, the seeds were sown and kept in the dark for 72 h at 25 ° C. Then, those seeds with an emerged radicle of 1.5 cm ± 0.2 cm were selected for further analysis.

### 2.5. Sowing in Different Substrates and Conditions

Sterile forest trays (Plásticos Solanas S.L., Zaragoza, Spain) were used, each composed of twelve 18-cm-high alveoli, a capacity of 300 cm$^3$, and a span of 5.3 cm × 5.3 cm. In total, forty-four trays were used, eleven for each type of substrate and treatment (four selected bacterial strains, their six mixtures and their respective controls without inoculum).

To avoid cross contamination, a single, pre-germinated seed was sown in each alveolus (emerged radicle approximately 3 cm). In each tray, a single strain (or mixture) and/or control was inoculated, so 12 replicates were tested for each condition. Table 3 shows the details of the loading procedure with substrates of the alveoli trays.

**Table 3.** Structuring of the load with substrates of each experimental group.

| Type of Substrate | Content of Each Alveolus |
|---|---|
| Contaminated soil, high [Hg] | Soil of "Plot 6". Capillary irrigation with sterile water, up to field capacity. |
| Contaminated soil, low [Hg] | Soil of "Plot 2". Capillary irrigation with sterile water, up to field capacity. |
| Vermiculite with Hg | 30 g of vermiculite. Capillary irrigation with 11 mL of 8 ppm HgCl$_2$ solution (2.6 mL/g), up to field capacity. |
| Vermiculite without [Hg] | 30 g of vermiculite. Capillary irrigation, up to field capacity. |

### 2.6. Inoculation with the Strains and Mixtures

Prior to inoculation, the strains were incubated in Nutritive Agar with 50 ppm of HgCl$_2$ for 24 h at 25 °C. Next, a Gram stain was carried out for the microscopic observation to verify the absence of contaminants. A bacterial suspension was prepared in 0.45% saline and adjusted to 0.5 McFarland (bacterial density 10$^8$ cfu/mL). A 0.45% saline solution was used (instead of 0.85%) to keep osmolarity. It was intended to avoid an increase in salinity, which could be aggravated by the incorporation of Hg salts and which could compromise the correct development of the seedlings. Each seed was inoculated with 1 mL of suspension.

### 2.7. Plant Growth Conditions

A phytotron equipped with white and yellow light was used, with a photoperiod of 11 h of light; Light intensity: 505 μmol m$^{-2}$ s$^{-1}$, stable temperature at 25 ± 3 °C. Irrigation was carried out every 48 h by capillarity with sterile tap water, with a volume, experimentally, of 350 mL/tray (12 alveoli).

*2.8. Harvest and Determination of Biometric Parameters*

Twenty days after sowing, we proceeded to harvest (aerial and root part). To determine the biometric parameters (weight and length), the root and aerial part were washed with distilled water. The rhizospheric fraction was preserved for use in future trials. With the freshly harvested plants, the following biometric parameters were measured: total weight (g), weight of the aerial part (g), weight of the root part (g), length of the aerial part (cm), length of the root part (cm), total number of leaves, and total number of secondary roots. To study the overall behavior of each part of the plant, the previous parameters were categorized into two groups of standardized measurements: "Aerial part" and "root part", both without units. Thus, the "aerial part" includes aerial weight, aerial length, and number of leaves. The "root part" includes root weight, root length and the number of roots.

*2.9. Statistical Analysis*

For the statistical analysis, SPSS v.26.0 program (Version 26.0 IBM Corp, Armonk, NY, USA) was used. First, one ANOVA test was carried out to analyze the behavior of the plant in the presence of the PGPR strains in the vermiculite and soil substrates, regardless of whether they had been treated with Hg. Next, the multiple comparison test known as "Least Significant Difference" (LSD) was performed in those parameters that displayed significance ($p$-value $\leq$ 0.05). This is a post-hoc analysis using "t" tests to perform all pairwise comparisons between group means. The objective of this test is to identify the strains and mixtures that produce significant variations in any of the biometric parameters studied, compared to the controls, in the presence of the two types of substrates tested.

Next, four ANOVA tests were performed, one for each type of treatment (vermiculite with/without Hg and soil with low/high [Hg]). In the parameters in which significant differences were obtained ($p$-value $\leq$ 0.05), an LSD test was performed to identify the strains (or mixtures) that promoted significantly higher growth of *Lupinus albus*, compared to the controls, including the variable Hg.

## 3. Results

In order to study the influence of the substrate (soil/vermiculite), regardless of the presence of Hg on the biometric variables, a first ANOVA was performed. The ANOVA didn't show significant differences between control and plants inoculated with the different strains and mixtures when the test was performed on vermiculite. However, when tested on the soil substrate, significant differences ($p$-value $\leq$ 0.05) were obtained for the variables designated as aerial length ($p$-value = 0.004), aerial weight ($p$-value = 0.036), and aerial part ($p$-value = 0.010). For the latter, LSD tests were performed to detect which strains and mixtures justified the significant differences. Results are shown in Figure 1.

The obtained results show that the B2 strain produced a significantly higher increase in the aerial weight of the plants compared to control (Figure 1A). Regarding the aerial length (Figure 1B), significantly higher growth was displayed compared to control in all strains and mixtures, except for CS1 and CS2 mixtures. The most effective strain is B2, while the best mixture was CS3 (A1, *Brevibacterium frigoritolerans* + B2, *Pseudomonas baetica*).

For the aerial part (Figure 1C), all strains and mixtures with the exception of CS1 and CS2 showed significant differences in terms of general aerial growth. This growth was higher than in control plants, with B2 strain being the most significant (*Pseudomonas baetica*), as well as CS6 mixture (A2, *Bacillus toyonensis* + B2, *Pseudomonas baetica*). Leaves number on the plants did not vary significantly according to the substrate in which the plant developed.

Based on the above, B2 strain, *Pseudomonas baetica*, was the most favorable to the development of the aerial part in those plants growing in soil substrate. This strain also participated in the mixtures that contributed the most to growth: CS3 (A1, *Brevibacterium frigoritolerans* + B2, *Pseudomonas baetica*), CS5 (B1, *Pseudomonas moraviensis* + B2, *Pseudomonas baetica*) and CS6 (A2, *Bacillus toyonensis* + B2, *Pseudomonas baetica*).

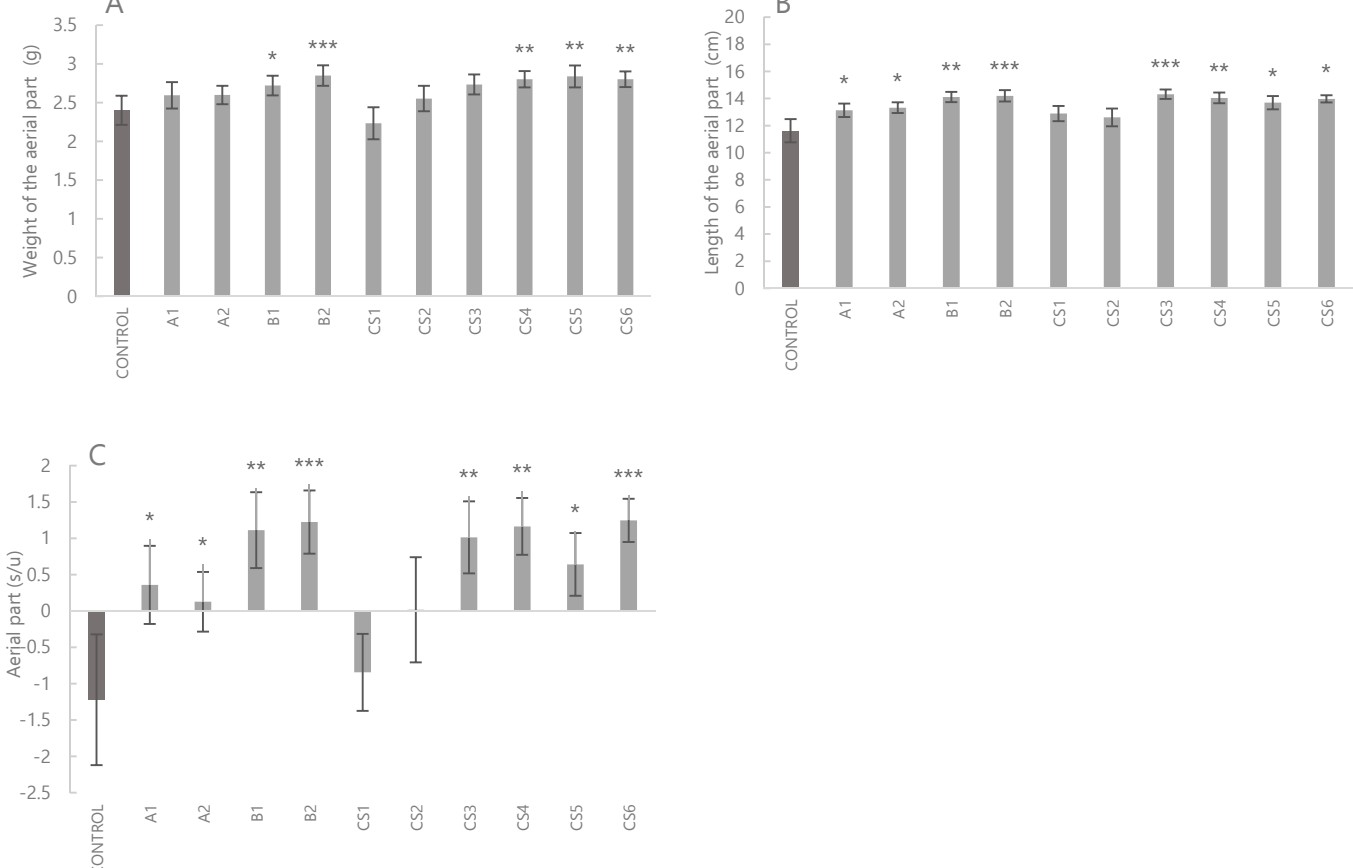

**Figure 1.** Results of the LSD test of the biometric parameters of plants grown in soil that showed significant differences in the ANOVA. (**A**): weight of the aerial part, (**B**): length of the aerial part, (**C**): "aerial part". The bars indicate standard error. * Indicate significant differences compared to the corresponding controls (*p*-value ≤ 0.05 and > 0.03). ** Indicate significant differences compared to the corresponding controls (*p*-value ≤ 0.03 and > 0.01). *** Indicate significant differences compared to the corresponding controls (*p*-value ≤ 0.01). s/u: no units. A1: *Brevibacterium frigolitolerans*, A2: *Bacillus toyonensis*, B1: *Pseudomonas moraviensis*, B2: *Pseudomonas baetica.* CS1 (A1 + B1), CS2 (A1 + A2), CS3 (A1 + B2), CS4 (B1 + A2), CS5 (B1 + B2), CS6 (A2 + B2).

In the second analysis, two ANOVAs were carried out to study how the presence of Hg influenced the plants' development, in each of the substrates separately (soil with high [Hg] and low [Hg]; and vermiculite with Hg and without Hg). For vermiculite, regardless of the presence of Hg or lack thereof, no significant growth differences were obtained in any of the parameters.

In soil with low [Hg], significant differences were obtained (*p*-value < 0.05) in the aerial weight, aerial length, and aerial part. To ascertain which strains/mixtures were responsible for the significant differences in the diverse biometric parameters in soils with low Hg concentration, an LSD test was performed. This analysis showed that B2 strain and CS6 mixture were contributing more significantly to the aerial development of the plant compared to control, especially in the increase in the weight of the aerial part (Figure 2A) and in the parameter "aerial part" (Figure 2C).

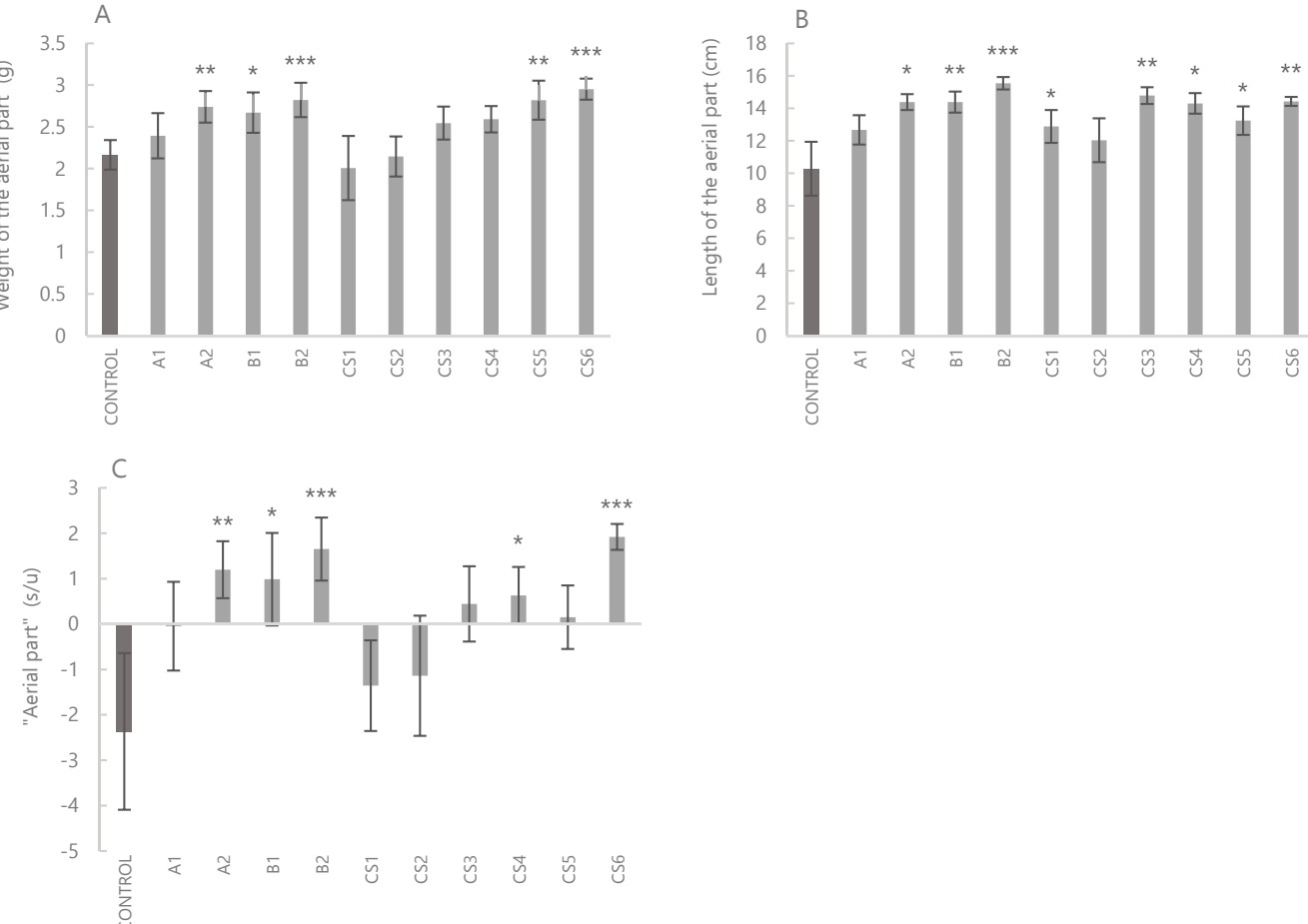

**Figure 2.** Results of the LSD analysis of the biometric parameters of the plants tested in low [Hg] for which significant differences were obtained based on the treatment (*p*-value ≤ 0.05). (**A**): weight of the aerial part, (**B**): Length of the aerial part, (**C**): "aerial part". The bars indicate standard error. * Indicate significant differences compared to the corresponding controls (*p*-value ≤ 0.05 and > 0.03). ** Indicate significant differences compared to the corresponding controls (*p*-value ≤ 0.03 and > 0.01). *** Indicate significant differences compared to the corresponding controls (*p*-value ≤ 0.01). s/u: no units. A1: *Brevibacterium frigolitolerans*, A2: *Bacillus toyonensis*, B1: *Pseudomonas moraviensis*, B2: *Pseudomonas baetica*. CS1 (A1 + B1), CS2 (A1 + A2), CS3 (A1 + B2), CS4 (B1 + A2), CS5 (B1 + B2), CS6 (A2 + B2).

In soil with high Hg concentration, significant differences were obtained (*p*-value < 0.05) for total weight, aerial part, aerial length, number of leaves, root part, root weight, and number of secondary roots (Figure 3). In this case, the CS3 mixture (A1, *Brevibacterium frigoritolerans* + B2, *Pseudomonas baetica*) is the one contributing most significantly to an increase in the weight of plants compared to controls (Figure 3A). In the root part, CS3 mixture (A1, *Brevibacterium frigoritolerans* + B2, *Pseudomonas baetica*) showed significant differences compared to controls in all biometric parameters. Noteworthy is the CS5 mixture (B1, *Pseudomonas moraviensis* + B2, *Pseudomonas baetica*), which provided greater growth in the root part parameters such as the number of roots (Figure 3–D). Strain B2 contributed stronger growth compared to control. Regarding aerial growth, the CS3 mixture (A1, *Brevibacterium frigoritolerans* + B2, *Pseudomonas baetica*) stood out once again in aerial part parameters and number of leaves (Figure 3F,G), while CS5 (B1, *Pseudomonas moraviensis* + B2, *Pseudomonas baetica*) significantly increased the growth of the aerial length compared to control (Figure 3E). Strains B1 and B2 also showed greater growth of the aerial part compared to control (Figure 3G).

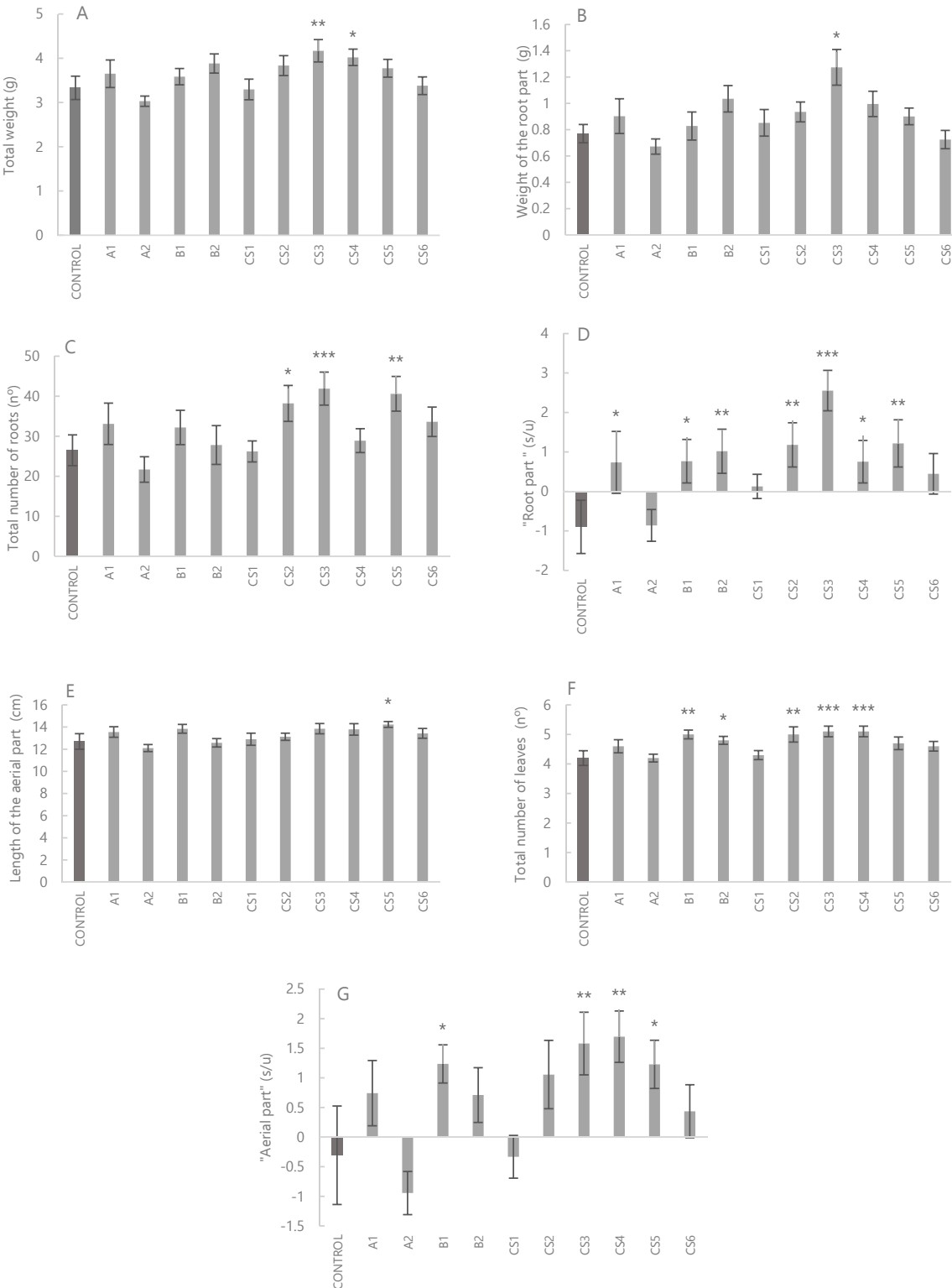

**Figure 3.** Results of LSD analysis of the biometric parameters of plants grown in soil with high [Hg], for which significant differences were obtained based on the treatment (*p*-value ≤ 0.05). (**A**): total weight, (**B**): weight of the root, (**C**): Total number of roots, (**D**): "Root part", (**E**): Length of the aerial part, (**F**): total number of leaves, (**G**): "aerial part". The bars indicate standard error. * Indicate significant differences compared to the corresponding controls (*p*-value ≤ 0.05 and > 0.03). ** Indicate significant differences compared to the corresponding controls (*p*-value ≤ 0.03 and > 0.01). *** Indicate significant differences compared to the corresponding controls (*p*-value ≤ 0.01). s/u: no units. A1: *Brevibacterium frigolitolerans*, A2: *Bacillus toyonensis*, B1: *Pseudomonas moraviensis*, B2: *Pseudomonas baetica*. CS1 (A1 + B1), CS2 (A1 + A2), CS3 (A1 + B2), CS4 (B1 + A2), CS5 (B1 + B2), CS6 (A2 + B2).

## 4. Discussion

In the present study we verified the effects on the growth of PGPR capabilities of a selection of bacterial strains (and their mixtures), in the first growth stages of *Lupinus albus* var. Orden Dorado plants by comparative analysis of their aerial biometric parameters and roots in different conditions of Hg contamination. Based on the results, a potential phytoremediation strategy based on a plant-bacteria symbiosis can be drawn up in order to offer an alternative use for soils subjected to high Hg contamination, such as those in Almadén.

Tested strains were selected based on BMRSI, which comprehensively assesses the degree of suitability of a strain for potential use in phytoremediation processes in the presence of heavy metals. In addition to the resistance to Hg, all the selected strains had PGPR capabilities, as indicated by BMRSI values greater than 6.5 [17]. PGPR strains promote plant growth on two levels: (i) directly, for example through the release of auxins or the production of ACC deaminase; and (ii) indirectly, through the synthesis of siderophores, among others. Thus, plants can develop biometrically and physiologically, even under abiotic stress by Hg thanks to the direct and indirect positive impact of PGPR. Rajkumar et al. [23] studied the same positive effect on legumes in the presence of heavy metals, once they had inoculated endophytic PGPR bacteria.

Studies have shown that the lupine *Lupinus albus* var. Orden Dorado, similar to other legumes, can extract heavy metals from soil [24], and is also tolerant to abiotic stressors, such as heavy metals presence and high salinity in soils. Therefore, it is a species widely used in bioremediation projects [25,26].

In the present study, to evaluate the effects of Hg on plant growth, the plants were grown with two different substrates (bulk soil and Vermiculite). Indeed, two types of bulk soil were used, obtained from the Almadén district (soil contaminated with Hg from "Plot 6", and a control soil with low Hg concentration, extracted from "Plot 2") [17]. The latter was used as control (instead of soil without Hg), to guarantee sample homogeneity and representativeness. To use a soil lacking Hg, a plot very far from the study site (Almadén) would have had to be sampled, which would have led to soil physicochemical changes of the soil and, therefore, to alterations in samples nature. With this assumption, it could not have been concluded whether changes in the response in the plant biometry would have been due to the conditions tested (inocula) or to the variation in soil properties. This justified the use of soil from an area that had the lowest known Hg concentration yet maintained the properties of the non-microbial fraction of the soil.

The second substrate used was vermiculite. Rodríguez et al. [27] have considered this inert substrate to be suitable for evaluating the effects of Hg on the strains (and mixtures) tested, without considering the shielding effect that soil produces, as this is a complex matrix with its own resident microbiome. Moreover, they demonstrated the ability of *Lupinus* spp to absorb and accumulate Hg in the stem when grown in vermiculite in the presence of Hg. According to López [28], even though there is no interaction between vermiculite and Hg, the root system of plants is capable of absorbing and accumulating heavy metal.

To examine and identify the strains and mixtures with the best phyto-rhizoremedial capability, a statistical analysis was carried out with regard to the way in which the diverse biometric variables studied vary in the different treatments. The first observation was that plants grew better in soil than in vermiculite, regardless of whether they had been treated with Hg or not. This may be related to the fact that as soil is a complex matrix, it allows for the contribution of nutrients and greater homeostasis, which favor plant growth. The development increased when the plants were also treated with the PGPR strains and mixtures. This finding is significant for further field trials. It suggests the existence of a high survival and colonization capacity of inoculated strains in a complex environment with a resident microbial community. Thus, it cannot be ruled out that together with the effect of the physicochemical factors inherent in the soil nature (e.g., the availability of mineral nutrients), the strains tested might have efficiently competed with the native

microorganisms. Finally, it has not been ruled out that synergistic processes might become established between the inoculated strains or mixtures and the native soil strains. However, these are only working hypotheses for a possible future experiment aimed at analyzing the survival and colonization of the root by the inocula.

Once the highest growth level of plants in soil with the presence of an inoculum was verified, the effect of Hg on plant development was analyzed. The effect of Hg on the development of the plant grown in vermiculite did not show significant differences in the biometry of the plant. Consequently, the authors did not delve deeper into this analysis.

The aerial length and aerial part parameters were significantly higher in the presence of an inoculum in the two conditions tested (in soil with low and high Hg concentration) (Figures 2 and 3E,F,G). On the other hand, plants growth in the presence of an inoculum in soil with high Hg concentration was higher compared to that of their respective controls in the root parameters (root length, root weight, and number of secondary roots). This plant organ is the main entry tissue for heavy metals in the plant, mainly by diffusion processes in the medium by means of a massive flow, and through cationic exchange. Negative charges of the rhizodermis cells interact with positive charges of heavy metals present in the soil, creating a dynamic balance that facilitates entry into the cell [29]. Among these three root parameters, the most significant growth was observed in the inoculum made with CS3 mixture (strain A1, *Brevibacterium frigotolerans* + strain B2, *Pseudomonas baetica*) (Figure 3B–D). Total weight, aerial part, and number of leaves of the plants sown in soil with a high concentration inoculated with CS3 mixture were also increased (Figure 3A,E,F).

Although the strains have individual PGPR characteristics, these can be increased or decreased when combined with mixtures. It depends on the interspecific competition, or synergy phenomena, which may result in specific behavior of the mixture exceeding what is expected from the sum of the parts, or is lower than the individual strains [30,31]. Therefore, it would be useful to study how the combination of rhizobacterial strains in the different mixtures tested can affect the different plant parameters. By analyzing the growth parameters in the presence of different inocula, it was observed that the most competitive mixture was CS3 (A1, *Brevibacterium frigotolerans* + B2, *Pseudomonas baetica*), as well as mixture CS5 (B1, *Pseudomonas moraviensis* + B2, *Pseudomonas baeitca*). These mixtures promoted significantly higher growth than control in tests carried out in soil with a high Hg concentration (Figure 3). On the other hand, the most competitive mixture in plants grown in soil with low Hg concentration was CS6, as well as CS5 to a lesser extent (Figure 2). Likewise, strains B1, *Pseudomonas moraviensis* and B2, *Pseudomonas baetica*, induced the highest growth values of all the biometric parameters, both with high and low Hg concentrations. To highlight that on an individual basis, strains B1 and B2 were those with the best PGPR capabilities and, in addition, they were part of those mixtures that proved to have higher PGPR capability in soils contaminated with Hg (CS3 and CS5).

Therefore, it could be stated that B1 and B2 strains, both individually and when forming mixtures, are the best candidates for further field phytorhizoremediation experiments on soils contaminated with Hg. Similar to the genus *Pseudomonas* in general, Strain B2, which is identified as *Pseudomonas baetica*, is characterized as a producer of auxins in concentrations greater than 5 μg/mL [32]. Auxins are plant growth regulators that are widely known to be very important, even though they were not the only ones. Strain B1, identified as *Pseudomonas moraviensis*, displays a tolerance of up to 100 μg/mg of Hg [17]. This strain stands out for production of IAA, the synthesis of siderophores, and the ability to hydrolyze ACC through the synthesis of ACC deaminase enzyme, which is related to the growth promotion of plants subject to environmental stress (pressure by Hg, for example).

Furthermore, the genus *Pseudomonas* is characterized by the production of siderophores that act as chelating agents to sequester iron (Fe). Fe is an element of the soil (generally present in its ionic form $Fe^{3+}$), usually scarce, and one of the main micronutrients necessary for plant development. Braud et al. [33] showed that the siderophores produced by the genus *Pseudomonas* have a greater affinity not only for $Fe^{3+}$ sequestration, but for other minerals as well, such as Hg. In the presence of Hg in the soil, the sequestration of $Fe^{3+}$

by siderophores is not negatively affected and remains available to plants and can also be incorporated thanks to the action of reducing enzymes that transform it into $Fe^{2+}$, another form that is soluble and bioavailable for the plant [34].

Strain A1 (*Brevibacterium frigoritolerans*) is also a producer of IAA, as well as siderophores. Consistent with the results obtained, other studies have already described the ability of this species to metabolize heavy metals, as well as to tolerate high concentrations of Hg [1,35].

Finally, A2 strain (*Bacillus toyonensis*) also favored the growth of *Lupinus albus*, although to a lesser extent, compared to the rest of the strains and mixtures. According to results obtained, and consistent with previous studies [36], this strain has interesting PGPR capabilities, such as the production of siderophores and IAA that make it suitable for its evaluation in future bioremediation uses.

## 5. Conclusions

1. Treatment of *Lupinus albus* var. Orden Dorado with PGPR strains (and mixtures) displays greater development among those plants grown in soil subjected to stress by Hg compared to those that have not received an inoculum.
2. The BMRSI appears to be a good indicator for the selection of PGPR strains for further use in phytorhizoremediation processes. Strains B1 (*Pseudomonas moraviensis*) and B2 (*Pseudomonas baetica*), used independently, are postulated as the best candidates for further in situ assays of phytorhizoremediation processes, as they promote higher growth than controls in all biometric parameters. B2 significantly increases "root part" parameter and total number of leaves" whereas B1, in addition, significantly promotes the increase in "aerial part".
3. The most promising mixtures for further testing are CS3 (A1, *Brevibacterium frigotolerans* + B2, *Pseudomonas baetica*), as it significantly increases almost all root and shoot biometric parameters and CS5 (B1, *Pseudomonas moraviensis* + B2, *Pseudomonas baeitca*), which significantly increases the number of roots and the aerial part length.

**Author Contributions:** Conceptualization, M.R., A.P. and P.A.J.; methodology, M.R., A.P., C.B., D.G. and P.A.J.; software, M.R.; validation, M.R.; formal analysis, M.R., A.P. and P.A.J.; investigation, M.R. and D.G.; re-sources, A.P. and P.A.J.; data curation, A.P.; writing—original draft preparation, M.R., C.B. and D.G.; writing—review and editing, M.R., A.P. and P.A.J.; visualization, M.R.; supervision, A.P. and P.A.J.; project administration, A.P.; funding acquisition, A.P. All authors have read and agreed to the published version of the manuscript.

**Funding:** This research was funded by FUNDACIÓN UNIVERSITARIA SAN PABLO CEU AND BANCO SANTANDER, grant number FUSP-BS-PPC01/2014.

**Institutional Review Board Statement:** Not applicable.

**Informed Consent Statement:** Not applicable.

**Data Availability Statement:** No new data were created or analyzed in this study. Data sharing is not applicable to this article.

**Acknowledgments:** Agradecemos al Centro de Investigaciones Científicas y Tecnológicas de Extremadura las facilidades ofrecidas para el acceso a las semillas de *Lupinus albus* var. Dorado.

**Conflicts of Interest:** The authors declare no conflict of interest.

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
