# Peer review of "Evaluation of the PGPR Capacity of Four Bacterial Strains and Their Mixtures, Tested on Lupinus albus var. Dorado Seedlings, for the Bioremediation of Mercury-Polluted Soils"

_processes, doi:10.3390/pr9081293_

Round 1

Reviewer 1 Report

In the present study, the authors investigated the plant growth promoting capability of four PGPR strains (and tandems) isolated from bulk soil and rhizosphere of plants grown in a soil district heavily contaminated with Hg in Spain. Biometric measurements were performed on root and aerial plants parts to identify the most suitable bacterial strains for further phyto-rhizoremediation studies.

The paper is well-structured and easy to read. The topic is interesting and addressed with a scientific sound approach. Methodology and results arejus appropriately detailed and the latters are discusses against relevant literature.

I have just minor corrections to the text which I made directly on the pdf.

Author Response

Dear Reviewer,

We want to thank you the time it has taken to review the manuscript as well as the value of all contributions.

We have attended to each one of them meticulously and we have attached the Word file with all the modifications (including those of other reviewers) in yellow

Thank you again and kind regards,

Reviewer 2 Report

The aim of the study was to attempt the PGPR bacterial strains inoculation in order to improve Lupinus albus seeds growth in soils contaminated with Hg. The biorecovery of contaminated soils is an important practical scientific issue. The great advantage of the manuscript is the fact that the bacterial strains were isolated by the Authors and that the starter cultures were designed. I have some minor questions and suggestions which should be discussed prior publication:

  1. Could you describe more precisely parameters which have been improved and which you underlined in the conclusions section?
  2. Could you please add the legend describing tandems on each figure not only in methodology section?
  3. Did you measure the decontamination yield in soil?
  4. What are the future perspectives of your study?

Author Response

Dear Reviewer,

We want to thank you the time it has taken to review the manuscript as well as the value of all contributions.

We have attended to each one of them meticulously and we have attached the Word file with all the modifications (including those of other reviewers) in yellow.

Additionally, below I give an answer to one of the questions raised and that I have not included in the manuscript:

3.  Did you measure the decontamination yield in soil?

4.  What are the future perspectives of your study?

Joint answer to both questions: The bioaccumulation of Hg has been analyzed under the test conditions. We have carried out a trial in which we analyzed the accumulation of this heavy metal in the root and stem under Hg stress conditions in the same plant species (L. albus var. Golden Order), in the presence of inocula with PGPR (Strains B1 and B2 for being the ones that have offered the most promising results, as analyzed) and absence (control). The test has been done using Almadén soil as a substrate. The results of the trial are promising and and they seem to point to a phytoprotective role of the strains by enzymatic reduction of Hg, which contributes indirectly (and additionally) to the growth and development of the plant.

For this reason, research has continued with the selected strains and the scope of the study has been extended to the characterization of their complete genome, which allows the phenotypic observations to be genetically justified (especially interesting the results concerning MerA gene over expression and plant growth promoting genes). These results will be the subject of another dissertation and future publication.

Please do not hesitate to send us any other consideration you may have. We will be happy to keep improving.

Thank you again and kind regards,

Marina
